# Monitoring of Nitrogen and Grain Protein Content in Winter Wheat Based on Sentinel-2A Data

**Haitao Zhao [1,2], Xiaoyu Song [1,*], Guijun Yang [1], Zhenhai Li [1], Dongyan Zhang [2] and Haikuan Feng [1]**

[1] Key Laboratory of Quantitative Remote Sensing in Agriculture of Ministry of Agriculture, Beijing Research Center for Information Technology in Agriculture, Beijing Academy of Agriculture and Forestry Sciences, Beijing 100097, China

[2] National Engineering Research Center for Agro-Ecological Big Data Analysis & Application, Anhui University, Hefei 230601, China

[*] Correspondence: songxy@nercita.org.cn; Tel.: +86-10-5150-3676

**Abstract:** Grain protein content (GPC) is an important indicator of wheat quality. Earlier estimation of wheat GPC based on remote sensing provided effective decision to adapt optimized strategies for grain harvest, which is of great significance for agricultural production. The objectives of this field study are: (i) To assess the ability of spectral vegetation indices (VIs) of Sentinel 2 data to detect the wheat nitrogen (N) attributes related to the grain quality of winter wheat production, and (ii) to examine the accuracy of wheat N status and GPC estimation models based on different VIs and wheat nitrogen parameters across Analytical Spectra Devices (ASD) and Unmanned Aerial Vehicle (UAV) hyper-spectral data-simulated sentinel data and the real Sentinel-2 data. In this study, four nitrogen parameters at the wheat anthesis stage, including plant nitrogen accumulation (PNA), plant nitrogen content (PNC), leaf nitrogen accumulation (LNA), and leaf nitrogen content (LNC), were evaluated for their relationship between spectral parameters and GPC. Then, a multivariate linear regression method was used to establish the wheat nitrogen and GPC estimation model through simulated Sentinel-2A VIs. The coefficients of determination ($R^2$) of four nitrogen parameter models were all greater than 0.7. The minimum $R^2$ of the prediction model of wheat GPC constructed by four nitrogen parameters combined with VIs was 0.428 and the highest $R^2$ was 0.467. The normalized root mean square error (nRMSE) of the four nitrogen estimation models ranged from 26.333% to 29.530% when verified by the ground-measured data collected from the Beijing suburbs, and the corresponding nRMSE for the GPC-predicted models ranged from 17.457% to 52.518%. The accuracy of the estimated model was verified by UAV hyper-spectral data which had resized to different spatial resolution collected from the National Experimental Station for Precision Agriculture. The normalized root mean square error (nRMSE) of the four nitrogen estimation models ranged from 16.9% to 37.8%, and the corresponding nRMSE for the GPC-predicted models ranged from 12.3% to 13.2%. The relevant models were also verified by Sentinel-2A data collected in 2018 while the minimum nRMSE for GPC invert model based on PNA was 7.89% and the maximum nRMSE of the GPC model based on LNC was 12.46% in Renqiu district, Hebei province. The nRMSE for the wheat nitrogen estimation model ranged from 23.200% to 42.790% for LNC and PNC. These data demonstrate that freely available Sentinel-2 imagery can be used as an important data source for wheat nutrition and grain quality monitoring.

**Keywords:** vegetation indices (VIs); plant nitrogen accumulation (PNA); plant nitrogen content (PNC); leaf nitrogen accumulation (LNA); leaf nitrogen content (LNC); multivariate linear regression

## 1. Introduction

As one of the world's major food crops, wheat has a huge market demand in China [1], and its production status directly affects the country's stability of agricultural product. The demand for high-quality food products has increased in recent decades in China [2] and grain protein content (GPC) is an important quality index for wheat [3]. Protein content above 12.5% in wheat provides sufficient gluten to form good dough for bread making, while wheat with protein content under 11% is suitable for making cookies. GPC is the main measurement of wheat quality and it is determined by the genetic background and, to a large extent, environmental factors, such as N supply, as well as water and temperature conditions [4–8]. Advanced site-specific knowledge of GPC will provide opportunities to adopt optimized strategies for grain harvesting [3]. Therefore, real-time monitoring of plant N status and a pre-harvest prediction of the grain and/or protein yield in wheat can assist producers in improving N management strategies, as well as in generating yield and quality maps [4].

The formation of grain protein is physically dependent on plant nitrogen accumulation and its translocation from leaves and shoots to the grains in the grain filling stage [3,9–14]. Most of the nitrogen that is converted into protein is taken up prior to anthesis, stored in the leaves, and remobilized during grain fill [15]. The plant may take up nitrogen both pre- and post-anthesis [14]. The greatest part of the nitrogen present in the harvest is assimilated pre-anthesis in the aboveground parts and is mobilized in the vegetative parts and redistributed to the grains [9–14]. Wang et al. [16] showed that the nitrogen content of winter wheat at the anthesis stage was indicative for the final grain protein content and the correlation coefficient between the leaf nitrogen concentration at anthesis and the grain protein content was 0.726 ($n = 26$). The nitrogen in the leaves is an important component of chlorophyll and the enzymes involved in photosynthesis [15]. Leaf nitrogen content (LNC) is an important indicator of the crop photosynthetic capacity [17] and is needed by agronomists for fertilizer recommendations [18]. Quantification of LNC can provide valuable information for monitoring crop physiology [19] and practicing precision farming [20] so as to improve the use and efficiency of nitrogen fertilizers.

Remote sensing has been widely used as a non-destructive approach for estimating the leaf N content in the past few decades [21–24]. Leaf N accumulation (LNA) can provide comprehensive information about leaf dry matter and LNC, thus reflecting leaf N status, as well as vegetation coverage during crop growth [25]. Meanwhile, plant N concentration (PNC) and accumulation (PNA) have also been used as indicators for assessing the plant N status for crops [26]. PNC, expressed on a land area basis, is the product of the plant N concentration and dry biomass. It can be used to indicate the N status of crops in the same growth stage [27]. PNA is highly variable within a single year and between years, sites, and crops, even when the N supplies from both the soil and additional fertilizer inputs are plentiful [28].

There are many studies regarding the estimation of the plant nitrogen status from canopy spectral reflectance data [29,30]. It is possible to predict GPC from remote sensing data if the remote sensing model of the plant nitrogen content is integrated with an agronomic model of the grain protein based on the plant nitrogen accumulation at the wheat anthesis stage [3]. Wang et al. [31] obtained a good inversion effect by constructing a GPC prediction model based on wheat canopy spectral parameters and LNC. Xue et al. [4] showed a strong relationship between the leaf N status and GPC, which indicates that canopy spectra can be used to predict GPC [32]. Huang et al. [33] found that the ratio of carotenoids to chlorophyll a in winter wheat leaves can be used as an intermediate variable to establish an inversion model between the spectral characteristics of wheat and the GPC.

As a means of rapid, non-destructive, and large-area simultaneous monitoring, satellite remote sensing technology has been proved to be useful for the inversion of various physiological and biochemical parameters of crops [34,35]. It is possible to use the remote sensing information to invert crop physiological and biochemical parameters and to monitor crop quality [36].

Satellite remote sensing enables growers to obtain spatially explicit information about crop conditions to make both within-season management decisions and post-season evaluations relating to nutrient or irrigation management zones [37]. The Landsat Thematic Mapper (TM) is the most

commonly used satellite platform for assessing the spatial variability of crop conditions, including biomass, leaf area, gross primary production, and yield [32,38–40]. Eitel et al. [41,42] simulated the broad-bands used by RapidEye from ground-based hyperspectral reflectance data to disentangle the contributions of wheat biochemistry (Chl, N) and structure (leaf area) in the prediction of N concentration. Perry et al. [43] compared measurements of ground-based vegetation indices (Vis) sensitive to N concentration (i.e., CCCI) using RapidEye imagery and found that the prediction of N concentration from satellites was confounded by inaccurate measurements of biomass and by challenges associated with scaling ground measurements to satellite pixels [32]. Liu et al. [44] constructed an inversion model between the vegetation index and winter wheat GPC using remote sensing images of the wheat anthesis and filling stages and obtained satisfactory inversion accuracy. Reyniers et al. [45] calculated the normalized vegetation index (NDVI) through the spectral parameters obtained by color infrared aerial images collected before wheat harvest and the Cropscan Spectrometer and established a model for predicting GPC quality. Li et al. (2012) established the GPC estimation model with multi-temporal Landsat TM/ETM data through a generalized regression neural network (GRNN) method [46].

Sentinel-2 is the latest generation of Earth observation satellites launched by the European Space Agency (ESA) in recent years. The Sentinel-2 mission consists of two satellites developed to support vegetation, land cover, and environmental monitoring. The Sentinel-2A satellite was launched by the ESA on 23 June 2015 and operates in a sun-synchronous orbit with a 10-day repeat cycle. A second identical satellite (Sentinel-2B) was launched on 7 March 2017. The Sentinel-2 MultiSpectral Instrument (MSI) acquires 13 spectral bands ranging from visible and near-infrared (VNIR) to shortwave-infrared (SWIR) wavelengths along a 290-km orbital swath. The spatial resolution of the four bands of blue, green, red, and near-infrared is 10 meters. Among the multispectral optical satellite data, Sentinel 2A/2B data is the only available data with three bands in the red-edge range, which is very effective for detecting vegetation information. Clevers et al. [47] have proved that these three red-edge bands are particularly suitable for estimating canopy chlorophyll and nitrogen (N) content.

The aim of this work is to investigate whether using VIs derived from Sentinel-2A/2B (with an emphasis on the red-edge band) can be used to detect the wheat N status and, furthermore, to quantitatively forecast the wheat grain protein of crops before they fully ripen. The objectives of this study are: (1) To determine the predictive capability of commonly used VIs collected during wheat "flowering/anthesis" stages to estimate N status in wheat using simulated Sentinel-2 broad-bands VIs from ground-based hyperspectral data; and (2) to evaluate the performance of a wheat GPC detection model based on different N parameters (PNA, PNC, LNA, and LNC) across a range of years, farms, and growing conditions and provide county-scale maps of wheat N parameters and GPC distribution for anthesis seasons (2017–2018) and to examine the model's precision.

## 2. Materials and Methods

### 2.1. Experimental Design

This study was conducted across three location representatives of the middle-high precipitation zone (600 mm annually) in the rainfed wheat-maize rotation region of Northern China. The 2003–2006 winter wheat experiments were carried out in the suburbs of Beijing; the 2013–2015 experiments were carried out in the National Experimental Station for Precision Agriculture in Beijing; and the 2017–2018 experiment was carried out in Renqiu, Hebei Province, south of Beijing (Figure 1). The summary for experiments and data collected in this study are listed in Table 1.

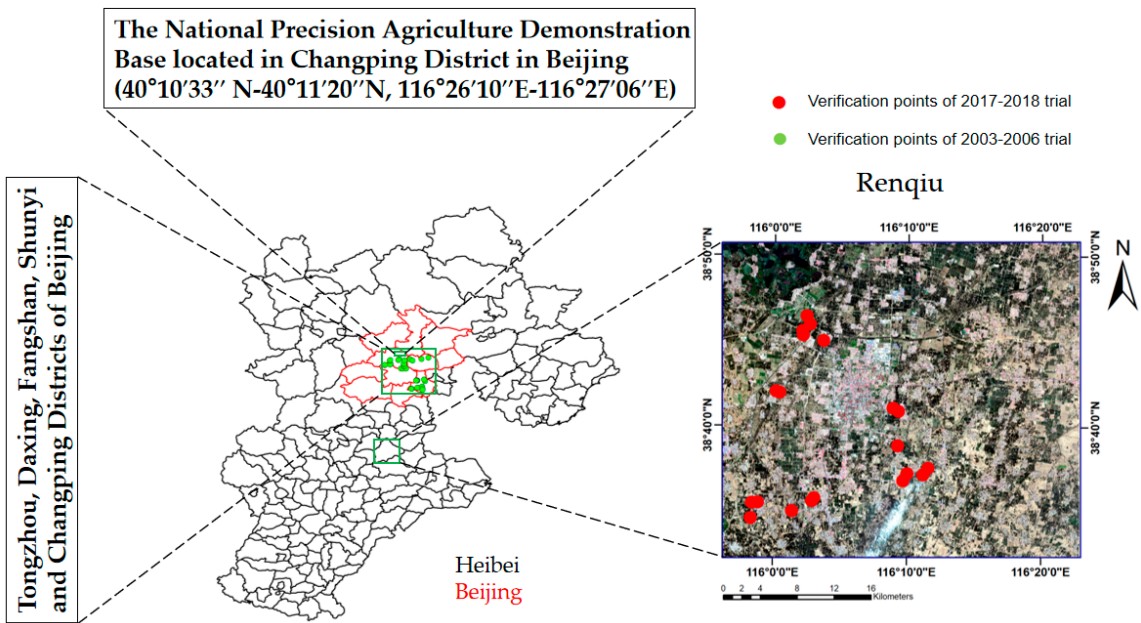

**Figure 1.** Location of the study area.

**Table 1.** Summary of experiment designs and ground-sampling data used in this study.

| Location | Trial | Year | Growth Stage | Treatment | Number of Samples | Data Collection | Model |
|---|---|---|---|---|---|---|---|
| National Experimental Station for Precision Agriculture | Trial 1 | 11 May 2013 | Anthesis | Nitrogen treatments | 32 | Canopy spectral data, Plant N, AGB, GPC | Calculation |
| | Trial 2 | 22 May 2014 | Anthesis | Nitrogen treatments | 48 | | |
| | Trial 3 | 13 May 2015 | Anthesis | Nitrogen treatments | 48 | UAV-UHD image, plant AGB, N, GPC | |
| Beijing suburb | Trial 4 | 17 May 2003 | Anthesis | Uniform Management | 21 | Canopy spectral data, Plant AGB, N, GPC | Validation |
| | Trial 5 | 18 May 2004 | Anthesis | Uniform Management | 25 | | |
| | Trial 6 | 8 May 2005 | Anthesis | Uniform Management | 14 | | |
| | Trial 7 | 10 May 2006 | Anthesis | Uniform Management | 13 | | |
| Renqiu | Trial 8 | 10 May 2018 | Anthesis | Uniform Management | 20 | Sentinel-2 data, Plant AGB, N, GPC | Validation |

Note: AGB: above ground biomass, N: nitrogen, GPC: grain protein content.

The climate characteristics of our study area are a typical warm temperate semi-humid monsoon continental climate, with high temperatures and rain in summer, cold and dry in winter, and short duration in spring and autumn. The annual frost-free stage is 180–200 days and the annual average temperature is 8–12 °C. The annual average precipitation spatial distribution is uneven (from 500 to 700 mm) and the rainfall season distribution is also very uneven as the summer precipitation accounts for about 80% of the annual precipitation and is mainly concentrated in June, July, and August. The average annual sunshine hours are between 2000 and 2800 h.

The National Experimental Station for Precision Agriculture is located in Xiaotangshan Town, Changping District, Beijing (40°10′33″ N–40°11′20″ N, 116°26′10″ E–116°27′06″ E). Trial 1, Trial 2, and Trial 3 were conducted in this station during 2013–2015.

Trial 1: The 2012–2013 trial was a randomized block with two replicates. The nitrogen treatments in the 2012–2013 growing season were 0 kg N per hectare, 104 kg N per hectare, 208 kg N per hectare, and 416 kg N per hectare. Nongda 211, Zhongmai 175, Zhongyou 206, and Jing 9843 were chosen as the

test cultivars. The plot area was 90 m$^2$ (10 m × 9 m). The wheat sowing date was 28 September 2012. In addition to the nitrogen applied to each treatment, the fertilizer level was 60 kg/ha P$_2$O$_5$ and 76.5 kg/ha K$_2$O, and the irrigation amount was 187 mm. The data obtained from this experiment were used for crop N status and GPC modeling.

Trial 2: The 2013–2014 trial was an orthogonal experimental design, with three factors including N rates, varieties, and irrigation volume over 2013–2014. The winter wheat varieties were Jing 9843 and Zhongmai 175. Four nitrogen levels (0, 90, 180, and 360 kg N per hectare) were used. The irrigation volume levels were 25, 171, and 317 mm. The plot area was 150 m$^2$ (10 m × 15 m) with three replicates for a total of 48 cells. The sowing date was 4 October 2013. In addition to the nitrogen applied to each treatment, the fertilizer level was 60 kg/ha P$_2$O$_5$ and 76.5 kg/ha K$_2$O, and the irrigation amount was 187 mm. The other field management parameters were the same as the local standard practices. The data obtained from this experiment was used for crop N status and GPC modeling.

Trial 3: At 2014–2015 trial, two winter wheat varieties, J9843 and ZM175 were selected for ground measurements during the flowering periods of 2015. Winter wheat was grown on 48 plots and four amounts of nitrogen fertilizer (N0 is no fertilizer, N1 is 195 kg/ha, N2 is 390 kg/ha, and N3 is 780 kg/ha) and three irrigation levels (W0 is rainfall only, W1 is rainfall plus 100 mm, and W2 is rainfall plus 200 mm) were applied in this trail. The data obtained from this experiment was used to evaluate the scale effects for winter N status estimation modeling used in this study.

Trial 4 to Trial 7 were conducted in Beijing suburbs from 2003 to 2006. The experiments were carried out in Tongzhou, Daxing, Fangshan, Shunyi, and Changping counties of Beijing. For each year, 10–30 wheat farmlands were selected as study fields. The geographical locations for these fields ranged from 115°25′ to 117°30′ in the east longitude and 39°38′ to 40°51′ in the north latitude (Figure 1). The farmlands selected in this study were flat and uniform in wheat growth, the area for each field was more than 5 ha. The farmlands were cultivated by farmers and managed by uniform fertilizer and water without special treatment. Remote sensing monitoring experiments were conducted at the wheat anthesis stage (17 May 2003; 18 May 2004; 8 May 2005; 10 May 2006). During each field trial, winter wheat in the middle of the field was selected as the sampling subplot. Plant samples were taken almost synchronously with the spectral measurements. Plant samples were immediately sealed in plastic bags and transported to the laboratory for subsequent analysis. All sampling locations were positioned using a handheld GPS. The data obtained from these experiments were used for model validation.

The 2017–2018 trial was carried out in the Renqiu area, which is located in the Cangzhou city of Hebei Province (38°32′17″ N–38°50′50″ N, 115°55′59″ E–116°22′55″ E). During the wheat anthesis stage in 2018, from 5 May to 10 May, 20 wheat fields whose area was greater than 10 ha were selected in the Renqiu area for ground investigation. In the middle of each trial field, plants with uniform growth were taken for four rows with length of 60 cm. All plant samples were uprooted and placed in the sample bag, sealed, and sent to the laboratory for further processing. All sampling positions were located with a handheld GPS.

## 2.2. Experimental Data Acquisition

### 2.2.1. Canopy Spectral Measurement

Determination of the canopy spectra of wheat was carried out during the wheat anthesis stage (17 May 2003; 18 May 2004; 8 May 2005; 10 May 2006; 22 May 2013; 7 May 2014). The ground hyperspectral uses the Fieldspec FR2500 field spectral emission spectrometer (ASD, Boulder, CO, USA) with a spectral range of 350–2500 nm, a spectral resolution of 1.4 nm at 350–1000 nm and 2 nm at 1000–2500 nm, and the spectral resampling interval is 1 nm. When measuring, the probe is measured perpendicular to the top of the canopy about 1.3 m. Calibration was performed before and after the measurement using a calibration plate. Each cell was measured 20 times when the weather was sunny and cloudless between 10:00 and 14:00, and the average was taken as the canopy spectrum of the treatment. In this study, the wheat canopy hyperspectral data from Trial 1 to 2, Trial 4 to 7 was

all convolved to the spectral band configuration of Sentinel-2 using the expected spectral response function of Sentinel-2 [48] through ENVI (The Environment for Visualizing Images) software, in order to evaluate the spectral response of wheat nitrogen in each Sentinel-2 channel.

### 2.2.2. Winter Wheat LAI and AGB Data

After canopy spectral measurements were completed, samples were collected for the determination of wheat leaf area index (LAI), wheat leaves, stems, ears, as well as wheat above ground biomass (AGB). In this study, wheat plants from a $40 \times 50$ cm subplot within each field were cut with scissors, then placed in a plastic bag and transported to the laboratory for subsequent analysis. All leaves, stems, and ears from 20 wheat plants in each plot were removed, put into a paper bag and dried at 80 °C to remove moisture. Once the sample weight became constant (about 24 h), they were weighed using a balance accurate to 0.001 g. Finally, the biomass per unit area was calculated based on the measured planting density and the dry weight of the samples. Winter wheat LAI and above ground biomass (AGB) for leaves (LAGB), stems (SAGB), and ears (EAGB) was calculated using:

$$ \text{LAI} = \frac{\text{Sa} \times \text{n}}{\text{p}} \tag{1} $$

$$ \text{LAGB} = \frac{\text{m}_\text{L} \times \text{n}}{\text{p}} \tag{2} $$

$$ \text{SAGB} = \frac{\text{m}_\text{S} \times \text{n}}{\text{p}} \tag{3} $$

$$ \text{EAGB} = \frac{\text{m}_\text{E} \times \text{n}}{\text{p}} \tag{4} $$

$$ \text{AGB} = \frac{(\text{m}_\text{L} + \text{m}_\text{S} + \text{m}_\text{E)} \times \text{n}}{\text{p}} \tag{5} $$

where *Sa* is the total area of all the leaves for 20 wheat plants, *n* is the number of winter wheat plants per unit area, *p* is the number of selected winter wheat plants ($p = 20$ in this study), $m_L$ is the dry weight of the wheat leaf sample for 20 wheat plants, $m_S$ is the dry weight of the wheat stem sample for 20 wheat plants, $m_E$ is the dry weight of the wheat ear sample for 20 wheat plants.

### 2.2.3. Winter Wheat Nitrogen Parameters

After measurement of the AGB, the leaf, stem, and ear samples were separated, ground, and passed through a 40-mesh screen. Wheat leaf nitrogen content (LNC), stem nitrogen content (SNC), and ear nitrogen content (ENC) were determined using a Kjeldahl nitrogen analyzer B-339 (Buchi AG, Flawil, Switzerland). Then three other wheat nitrogen parameters, LNA, PNC, and PNA, which represent wheat leaf- and canopy-level nitrogen status, were calculated using Equations (6) to (8):

$$ \text{LNA} = \text{LAGB} \times \text{LNC} \tag{6} $$

$$ \text{PNA} = \text{LAGB} \times \text{LNC} + \text{SAGB} \times \text{SNC} + \text{EAGB} \times \text{ENC} \tag{7} $$

$$ \text{PNC} = \frac{\text{LAGB} \times \text{LNC} + \text{SAGB} \times \text{SNC} + \text{EAGB} \times \text{ENC}}{(\text{LAGB} + \text{SAGB} + \text{EAGB})} \tag{8} $$

### 2.2.4. Winter Wheat GPC Data

For experiments conducted at the National Experimental Station for Precision Agriculture during 2012–2014, after the wheat matured, 1 m$^2$ of wheat was taken from each cell, dried, and threshed. Winter wheat's GPC was determined using an Infratec TM 1241 Near-Infrared Grain Analyzer (FOSS Inc., Denmark).

For experiments conducted in the Beijing suburbs during 2003–2006 and in Renqiu during 2017–2018, five representative 1-m$^2$ areas of wheat plant were taken from each field by hand after the wheat matured, and then the wheat grain was dried and threshed. The winter wheat's GPC was determined by an Infratec TM 1241 near-infrared grain analyzer.

### 2.2.5. Acquisition and Processing of Unmanned Aerial Vehicle (UAV) Remote-Sensing Images

An UAV sensor platform, DJI S1000 UAV (SZ DJI Technology Co., Ltd., Sham Chun, China) with eight propellers, which is very stable at low flight speed and low altitude, was used in Trail 3 in this study. The UHD 185 Firefly (UHD 185 firefly, Cubert GmbH, Ulm, Baden-Württemberg, Germany) is a snapshot hyperspectral sensor. The UHD 185 has a short exposure and integration time, weighs 0.47 kg, and measures $195 \times 67 \times 60$ mm$^3$. Its operating range spans from the visible to the near-infrared (wavelength range: 450 nm to 950 nm, 8 nm @ 532 nm). Hyperspectral data cubes were automatically resampled to 4 nm spacing. Collected radiation is recorded as a $1000 \times 1000$ (1 band) panchromatic image and a $50 \times 50$ (125 bands) hyperspectral cube. The fusion steps are implemented in Cubert Cube-Pilot software (Cube-Pilot, Version 1.4, Cubert GmbH, Ulm, Baden-Württemberg, Germany). After fusion, all hyperspectral images with $1000 \times 1000$ (125 bands) were stitched together using an image stitching process, and the final result is shown in Figure 2.

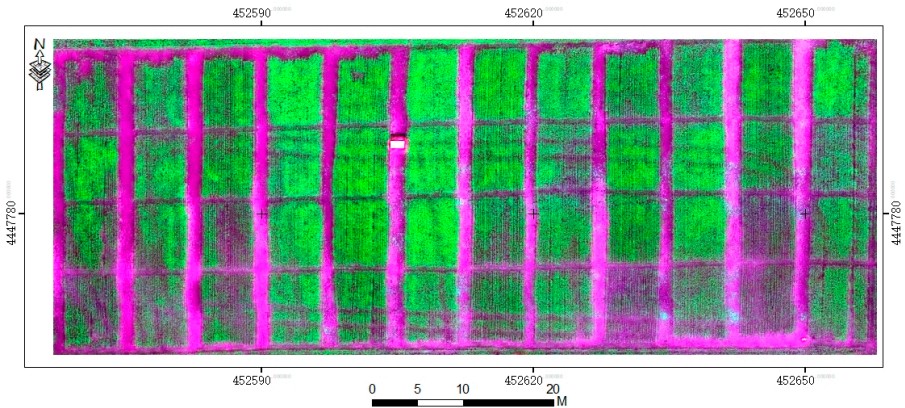

**Figure 2.** UAV hyper-spectral image for Trail 3.

Flights were conducted during the wheat anthesis stage (13 May 2015). The UAV flight altitude is 50 m and the spatial resolution of UAV-UHD image is 0.01 m. The UHD 185 was calibrated on the ground before the UAV flight by using Cubert Cube-Pilot software (Version 1.4) and a BaSO4 whiteboard. The original DN values of the UAV-DC images were calibrated by imaging a black-and-white fabric placed on the ground and using

$$DN_i = \frac{DN_{original} - DN_{black}}{DN_{white} - DN_{black}} \times 255 \tag{9}$$

where $DN_i$ is the band names, such as R, G, B, $DN_{original}$ is the original DN value of the high-definition digital camera images; and $DN_{original}$ and $DN_{black}$ are the original DN values from the white-and-black fabric.

In this study, the UAV hyperspectral image from Trial 3 was first convolved to the spectral band configuration of Sentinel-2 using the expected spectral response function of Sentinel-2 [48] through ENVI software. Then, the original 0.02 m simulated sentinel-2 image was resized to 0.5 m to 2.5 m spatial resolution images using the ENVI Resize tool in this study in order to estimate the wheat N and GPC estimation model accuracy.

### 2.2.6. Sentinel-2 Satellite Data Acquisition and Preprocessing

The Sentinel-2A/2B satellite data used in this study were downloaded from the official website of the ESA (https://scihub.copernicus.eu/dhus/#/home). Detailed information about the Sentinel-2 MSI is listed in Table 2.

**Table 2.** Sentinel 2A/2B MultiSpectral Instrument (MSI) data band information.

| Sentinel-2 Bands | Central Wavelength (nm) | Spatial Resolution (m) |
|---|---|---|
| Band 1—Coastal aerosol | $443 \pm 10$ | 60 |
| Band 2—Blue | $490 \pm 32.5$ | 10 |
| Band 3—Green | $560 \pm 17.5$ | 10 |
| Band 4—Red | $665 \pm 15$ | 10 |
| Band 5—Vegetation red-edge | $705 \pm 7.5$ | 20 |
| Band 6—Vegetation red-edge | $740 \pm 7.5$ | 20 |
| Band 7—Vegetation red-edge | $783 \pm 10$ | 20 |
| Band 8—NIR [1] | $842 \pm 57.5$ | 10 |
| Band 8A—Vegetation red-edge | $865 \pm 10$ | 20 |
| Band 9—Water vapor | $945 \pm 10$ | 60 |
| Band 10—SWIR [2]-Cirrus | $1375 \pm 15$ | 60 |
| Band 11—SWIR | $1610 \pm 15$ | 20 |
| Band 12—SWIR | $2190 \pm 90$ | 20 |

Note: [1] near-infrared; [2] shortwave-infrared.

A total of seven scenes of Sentinel-2A/2B images were obtained in this experiment (Table 3). The radiometric calibration and atmospheric correction of Sentinel-2A/2B in this study was completed using the plugin Sen2Cor-02.05.05 in the SNAP software. Sentinel band 5 to band 7, band 11, and band 12 data were resampled to 10-m spatial resolution using the SNAP software and then exported to the ENVI format. The follow-up work, such as wheat area extraction and VIs calculation, were all completed in ENVI5.3 [49]. The Sentinel-2B image of 8 May 2018, which was collected in the wheat anthesis season, was used as the data source for wheat nitrogen monitoring and GPC prediction. The other six scenes, which cover the winter wheat greening stage to the harvest stage in the Renqiu area, were used to extract the winter wheat planting area in 2018.

**Table 3.** Acquired remote sensing imagery types and time, as well as the winter wheat growth season.

| Image Acquisition Date | Image Type | Wheat Growth Season | Data Usage |
|---|---|---|---|
| 2018-03-01 | Sentinel-2A | Greening | Classification |
| 2018-03-16 | Sentinel-2B | Up | Classification |
| 2018-04-20 | Sentinel-2A | Flag | Classification |
| 2018-05-08 | Sentinel-2B | Anthesis | Nitrogen monitoring |
| 2018-05-30 | Sentinel-2A | Filling | Classification |
| 2018-06-04 | Sentinel-2B | Milking | Classification |
| 2018-06-14 | Sentinel-2B | Harvesting | Classification |

A supervised classification method of maximum likelihood combined with decision tree classification was used to complete the extraction of the winter wheat planting area. The wheat extraction results are shown in Figure 3.

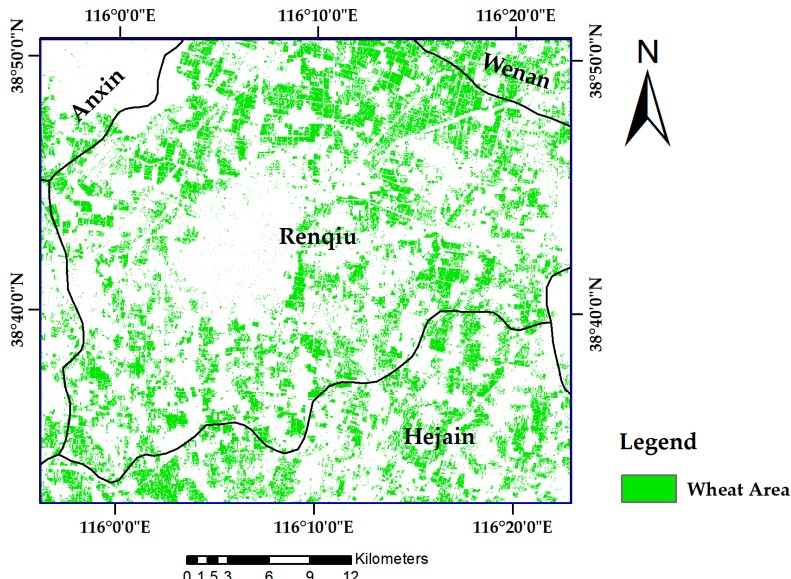

**Figure 3.** Winter wheat planting area in Renqiu area in 2018. The green part is the wheat growing area.

## 2.3. Method

This paper aims to investigate the feasibility of using VIs derived from Sentinel-2A/2B (with an emphasis on the red-edge band) to detect the wheat N status and, furthermore, to quantitatively forecast the wheat grain protein of crops before they fully ripen. First, we simulated the Sentinel-2A/2B multispectral data from wheat canopy hyperspectral data for wheat nitrogen and GPC monitoring. Then, the relationship between different spectral parameters and wheat nitrogen index and wheat grain protein content was analyzed. The wheat nitrogen estimation models and GPC estimation model were then established based on the selected wheat nitrogen-sensitive spectral VIs. The estimation accuracy for the models was verified by simulated Sentinel-2 data obtained from UAV-UHD image in Trail 3 and ASD spectral data in the 2003–2006 experiment in Beijing. Next, the models were applied in the Renqiu area and verified by the real Sentinel-2A data and the ground measured data. Figure 4 shows the workflow of the study.

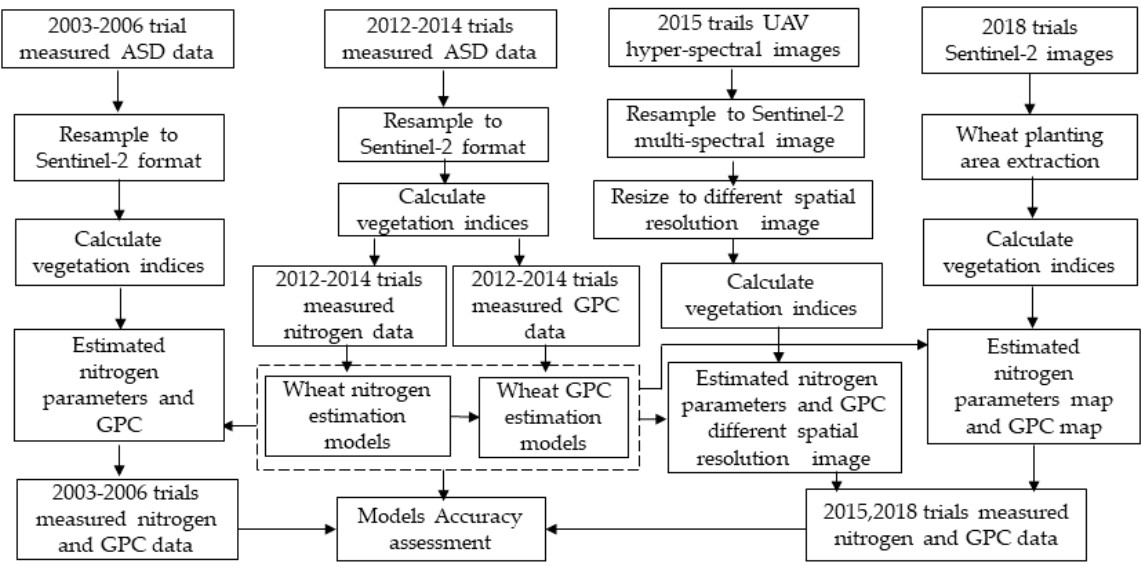

**Figure 4.** Workflow of the study.

### 2.3.1. Vegetation Index Selection

According to the spectral characteristics of winter wheat and the literature [50,51] and taking full advantage of the three red-edge bands carried by the Sentinel-2A/AB data, 14 vegetation indices were selected for estimating the wheat N status (Table 4).

**Table 4.** Calculation formulas of the vegetation indices.

| Vegetation Index | Name | Formula | References |
|---|---|---|---|
| MTCI | MERIS Terrestrial Chlorophyll Index | $(R_{VRE2} - R_{VRE1})/(R_{VRE1} - R_{Red})$ | [52] |
| RDVI$_{705}$ | Re-normalized Difference Vegetation Index | $(R_{VRE2} - R_{VRE1})/(R_{VRE1} - R_{Red})$ | [53] |
| mSR | Modified Simple Ratio | $(R_{VRE2} - R_{VRE1})/(R_{VRE1} - R_{Red})$ | [54] |
| mSR2 | mSR$_{[705,750]}$ | $(R_{VRE2} - R_{VRE1})/(R_{VRE1} - R_{Red})$ | [55] |
| SR$_{705}$ | Simple Ratio$_{[705,750]}$ | $R_{VRE2}/R_{VRE1}$ | [56] |
| MCARI | Modified Chlorophyll Absorption Ratio Index | $((R_{VRE1} - R_{Red}) - 0.2(R_{VRE1} - R_{Green}))(R_{VRE1}/R_{Red})$ | [57] |
| MCARI$_{[705,750]}$ | MCARI$_{[705,750]}$ | $((R_{VRE2} - R_{VRE1}) - 0.2(R_{VRE2} - R_{Green}))(R_{VRE2}/R_{VRE1})$ | [58] |
| REIP$_{S2}$ | Red-Edge Inflection Position of Sentinel-2 bands | $705 + 35 \times \{[0.5(R_{VRE3} + R_{Red}) - R_{VRE1}]/(R_{VRE2} - R_{VRE1})\}$ | [59] |
| CI$_{RE1}$ | Red-edge Chlorophyll Index | $(R_{NIR}/R_{VRE1}) - 1$ | [57] |
| IREC | Inverted Red-Edge Chlorophyll Index | $(R_{VRE3} - R_{RED}) \times R_{VRE2}/R_{VRE1}$ | [60] |
| RED EDGE NDVI | Red-edge NDVI | $(R_{VRE3} - R_{RED}) \times R_{VRE2}/R_{VRE1}$ | [61] |
| GSR | green SR | $R_{NIR}/R_{Green}$ | [62] |
| CRI 1 | Carotenoid Reflectance Index 1 | $(1/R_{Blue}) - (1/R_{Green})$ | [63] |
| CVI | Chlorophyll Vegetation Index | $(R_{NIR}/R_{Green}) \times (R_{Red}/R_{Green})$ | [64] |

### 2.3.2. Multiple Linear Regression Model

The relationship between the spectral parameters and winter wheat N status in the anthesis stage was analyzed, followed by the analysis of the relationship between the grain proteins, wheat N nitrogen indicators, and spectral parameters. A multivariate linear regression (MLR) algorithm with fast modeling and no complicated calculation was applied to detect the wheat nitrogen status and the wheat GPC [65]. The multiple linear regression models for wheat's four nitrogen parameters (LNC, LNA, PNC, PNA) and GPC were established based on the simulated Sentinel-2 VIs from Trial 1 and Trial 2 data. A total of 80 experimental data were obtained in the 2012–2014 experiment, three-quarters of which were randomly selected ($n$ = 60) for modeling, and the remaining quarter of the samples ($n$ = 20) was used for verification.

This study mainly uses the Tensor Flow framework to achieve regression and prediction and combines the Stochastic Gradient Descent algorithm to train the model. The square loss function is used as the loss function in the training. As shown in Equation (10), where $y_i$ represents the observed value, $y_i^{pred}$ represents the predicted value, and $n$ is the number of training samples:

$$L = \sum_{i}^{n} (y_i^{pred} - y_i)^2 \qquad (10)$$

### 2.3.3. Accuracy Verification

The MLR models' estimated accuracy for the N parameter and GPC were evaluated by the simulated Sentinel-2 VIs from Trial 3 to Trial 6 and then by the real Sentinel-2B data from Trial 7. The relevant model accuracy indicators were evaluated by the normalized root mean squared error (nRMSE) [66–69]. When describing the accuracy of the verification model using nRMSE, a significant difference limit is generally given; for example, the accuracy is considered excellent when nRMSE < 10%, good if 10% ≤ nRMSE < 20%, acceptable if 20% ≤ nRMSE < 30%, and poor if nRMSE ≥ 30% [69].

## 3. Results

### 3.1. The Distribution of Experimentally Measured Data

We obtained the measured data of LNC, LNA, PNC, PNA, and GPC for a total of seven trials in three experimental regions. The 2012–2014 experiments in the National Experimental Station for Precision Agriculture in Beijing obtained 80 sets of measured data for calculation, the 2003–2006 winter wheat experiments in the suburbs of Beijing obtained 63 sets of measured data for verification, and the 2017–2018 experiment in the Renqiu area obtained 20 sets of data for verification. The distribution of the experimental data between the three geographical spans was analyzed using a box plot (Figure 5).

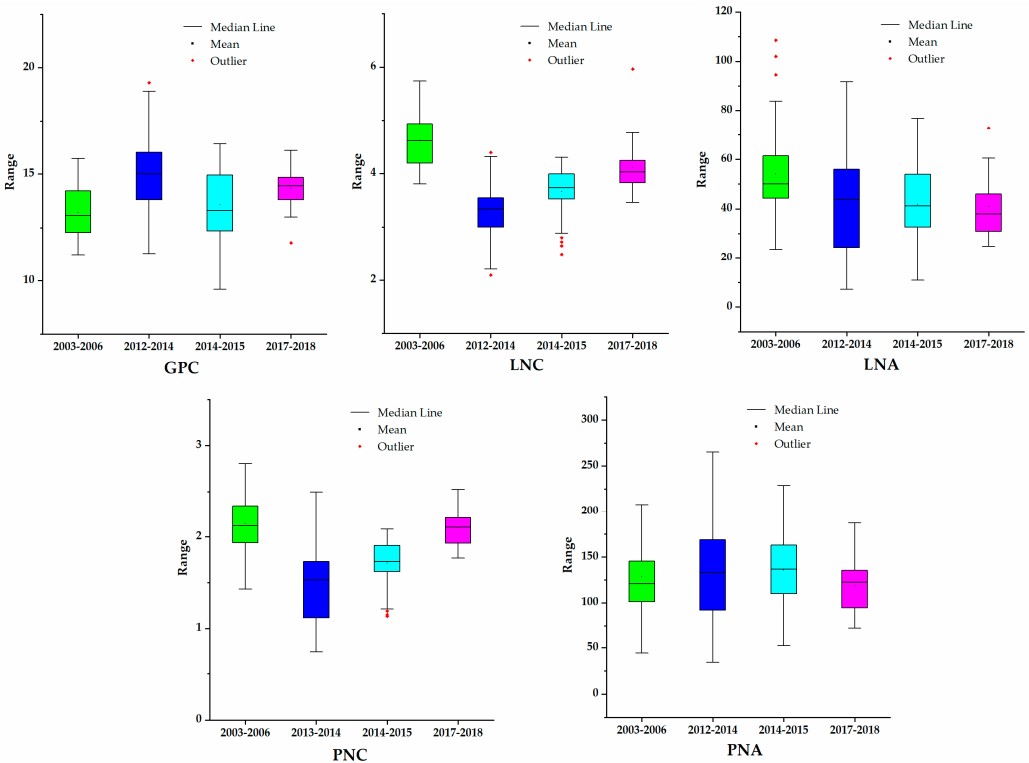

**Figure 5.** The distribution of the measured data.

### 3.2. Correlation Analysis between Simulated Sentinel-2A Vegetation Indices and Nitrogen Nutrition Parameters of Wheat at the Anthesis Stage

The relationship between simulated Sentinel-2 VIs and wheat plant nitrogen parameters and AGB for data collected from experiments carried out in the National Experiment Station for Precision Agriculture during 2012–2014 (Trial 1 and Trial 2) was analyzed and the results are listed in Table 5. The correlation between simulated Sentinel-2 VIs and above ground biomass (ABG), the four wheat nitrogen parameters (PNA, PNC, LNA, and LNC) all reached a significant correlation level (Table 5; $p < 0.01$).

The top five vegetation indices with the highest correlation coefficients were selected for each nitrogen nutrition index. The correlation between the selected five vegetation indices was determined, the vegetation indices with low correlation were removed to eliminate redundant information and, finally, two suitable vegetation indices were selected for nitrogen estimation.

According to the selection rules above, the vegetation indices selected to invert each nitrogen index are shown in Table 6. Figure 6 shows the scatter plot between four nitrogen nutrition parameters and their accordingly vegetation indices. There are good linear relationship between the nitrogen parameters and the selected vegetation indices.

**Table 5.** Correlation between vegetation indices (VIs) and above ground biomass (AGB), nitrogen nutrition parameters for Trial 1 and Trial 2 ($n = 80$).

| Vegetation Index | AGB (kg/hm$^2$) | PNA (kg/hm$^2$) | PNC (%) | LNA (kg/hm$^2$) | LNC (%) |
|---|---|---|---|---|---|
| MTCI | 0.713 ** | 0.883 ** | 0.875 ** | 0.836 ** | 0.883 ** |
| $RDVI_{705}$ | 0.780 ** | 0.881 ** | 0.858 ** | 0.856 ** | 0.839 ** |
| mSR2 | 0.765 ** | 0.872 ** | 0.867 ** | 0.851 ** | 0.840 ** |
| $SR_{705}$ | 0.738 ** | 0.882 ** | 0.884 ** | 0.855 ** | 0.848 ** |
| $MCARI_{[705,750]}$ | 0.731 ** | 0.888 ** | 0.879 ** | 0.856 ** | 0.830 ** |
| $REIP_{S2}$ | 0.753 ** | 0.873 ** | 0.866 ** | 0.846 ** | 0.893 ** |
| $CI_{RE1}$ | 0.711 ** | 0.891 ** | 0.895 ** | 0.856 ** | 0.862 ** |
| IREC | 0.717 ** | 0.899 ** | 0.893 ** | 0.863 ** | 0.844 ** |
| MCARI | −0.782 ** | −0.805 ** | −0.822 ** | −0.819 ** | −0.839 ** |
| RED EDGE NDVI | 0.733 ** | 0.886 ** | 0.880 ** | 0.853 ** | 0.881 ** |
| mSR | 0.760 ** | 0.867 ** | 0.871 ** | 0.857 ** | 0.822 ** |
| GSR | 0.719 ** | 0.880 ** | 0.894 ** | 0.857 ** | 0.860 ** |
| CVI | 0.607 ** | 0.700 ** | 0.765 ** | 0.743 ** | 0.693 ** |
| CRI 1 | 0.684 ** | 0.783 ** | 0.796 ** | 0.743 ** | 0.838 ** |

Note: ** r (0.01, 80) = 0.283, indicates significance at the 0.01 probability level; * r (0.05, 80) = 0.217, indicates significance at the 0.05 probability level.

**Table 6.** Vegetation indices selected for inversion of each nitrogen parameters.

| Nitrogen Nutrition Index | Vegetation Index |
|---|---|
| PNA | $SR_{705}$ $REIP_{S2}$ |
| PNC | MCARI $CI_{RE1}$ |
| LNA | mSR $CI_{RE1}$ |
| LNC | MTCI RED EDGE NDVI |

**Figure 6.** Relationship between nitrogen nutrition parameters and vegetation indices.

A total of four inversion models of wheat nitrogen parameters were then established by a multiple linear regression algorithm using the vegetation indices selected in Table 6. The models are shown in Table 7, and the modeling results are shown in Figure 7.

**Table 7.** Multiple regression models for different nitrogen nutrition parameters in the wheat anthesis stage.

| Parameters | Regression Model |
|---|---|
| PNA | $PNA_{Pred} = 23.537 \times SR_{705} + 6.178 \times REIP_{S2} - 4453.427$ |
| PNC | $PNC_{Pred} = 0.567 \times MCARI + 0.103 \times CI_{RE1} + 0.696$ |
| LNA | $LNA_{Pred} = 5.024 \times mSR + 4.962 \times CI_{RE1} + 1.924$ |
| LNC | $LNC_{Pred} = 0.179 \times MTCI + 3.093 \times RED EDGE NDVI + 1.804$ |

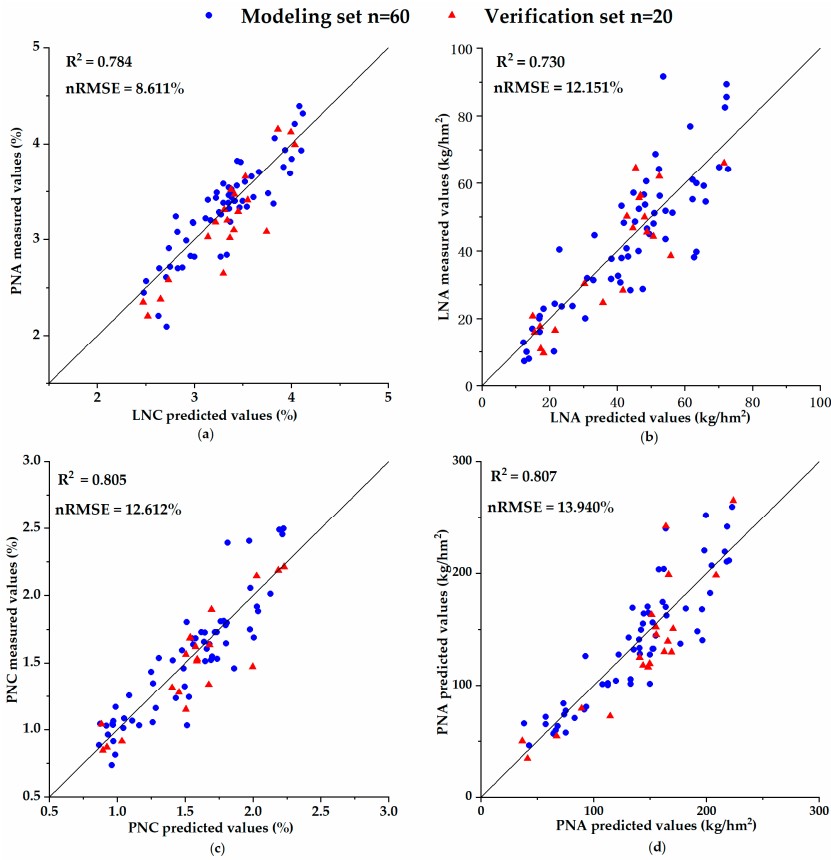

**Figure 7.** Relationship between predicted and measured values of nitrogen nutrition parameters. (**a**) Leaf nitrogen content (LNC) estimation results based on vegetation indices. (**b**) Leaf nitrogen accumulation (LNA) estimation results based on vegetation indices. (**c**) Plant nitrogen content (PNC) estimation results based on vegetation indices. (**d**) Plant nitrogen accumulation (PNA) estimation results based on vegetation indices.

Figure 7 Shows that for the four nitrogen nutrition parameters inversed by the vegetation indices, the maximum $R^2$ of the nd the minimum was for LNA (0.730). Modeling accuracy of all four models reached a very significant level. Therefore, the four nitrogen nutrition parameters could be inverted by the vegetation indices and the models can predict the nitrogen nutrition parameters well. To verify the indicator nRMSE based on the verification set, the minimum LNC is 8.611%, which is not significantly different. The nRMSE of LNA, PNC, and PNA were 12.151%, 12.612%, and 13.940%, respectively. Considering the accuracy verification indicators of each modeling set and verification set, PNA has a higher modeling accuracy $R^2$ and a smaller nRMSE and so the model is more stable.

### 3.3. Relationship between the Wheat Grain Protein Content and the Nitrogen Index, as well as Vegetation Indices

Most of the nitrogen that is taken up prior to flowering and stored in the leaves and remobilized during the grain filling stage [15]. It is possible to predict the wheat GPC from remote sensing data if the remote sensing model of the plant nitrogen content is integrated with an agronomic model of grain protein based on plant nitrogen accumulation at anthesis. The correlation coefficient between the four wheat nitrogen nutrition parameters and the winter wheat GPC in the anthesis stage all reached a very significant level ($p < 0.01$). Among them, LNC had the highest correlation with winter wheat GPC, and the correlation coefficient reached 0.660, while PNC reached 0.615, PNA 0.599, and LNA 0.483. Table 8 shows the relationship between the VIs and wheat GPC. It can be seen from Table 8 that all the correlation coefficients between vegetation indices and the winter wheat GPC were highly significant ($p < 0.01$). Among them, MTCI had the highest correlation with winter wheat GPC, followed by IREC.

**Table 8.** Correlation between spectral parameters and wheat GPC from Trial 1 and Trial 2 (*n* = 80).

| Vegetation Index | Correlation Coefficient | Vegetation Index | Correlation Coefficient |
|---|---|---|---|
| MTCI | 0.628 ** | IREC | 0.624 ** |
| $RDVI_{705}$ | 0.535 ** | MCARI | −0.429 ** |
| mSR2 | 0.515 ** | RED EDGE NDVI | 0.596 ** |
| $SR_{705}$ | 0.560 ** | mSR | 0.503 ** |
| $MCARI_{[705,750]}$ | 0.592 ** | GSR | 0.587 ** |
| $REIP_{S2}$ | 0.585 ** | CVI | 0.622 ** |
| $CI_{RE1}$ | 0.608 ** | CRI 1 | 0.284 ** |

Note: ** r (0.01, 80) = 0.283, indicates significance at the 0.01 probability level; * r (0.05, 80) = 0.217, indicates significance at the 0.05 probability level.

Considering the correlation between spectral parameters and wheat GPC listed in Table 9, the wheat GPC estimation models were constructed based on the four wheat nitrogen parameters (PNA, PNC, LNA, and LNC) and spectral VIs. The models are shown in Table 9. The correlations between the measured and predicted GPC values for the four models are shown in Figure 8.

**Table 9.** Multiple regression models of the wheat grain protein content.

| Parameters | Regression Model |
|---|---|
| $GPC_{PNA}$ | $GPC_{PNA} = 1.424 \times IREC + 0.461 \times CVI + 0.001 \times PNA_{Pred} + 10.536$ |
| $GPC_{PNC}$ | $GPC_{PNC} = 0.505 \times IREC + 0.685 \times CVI + 0.646 \times PNC_{Pred} + 9.980$ |
| $GPC_{LNA}$ | $GPC_{LNA} = 1.143 \times IREC + 0.664 \times CVI + 0.009 \times LNA_{Pred} + 9.700$ |
| $GPC_{LNC}$ | $GPC_{LNC} = 3.077 \times IREC + 1.871 \times CVI - 5.737 \times LNC_{Pred} + 20.503$ |

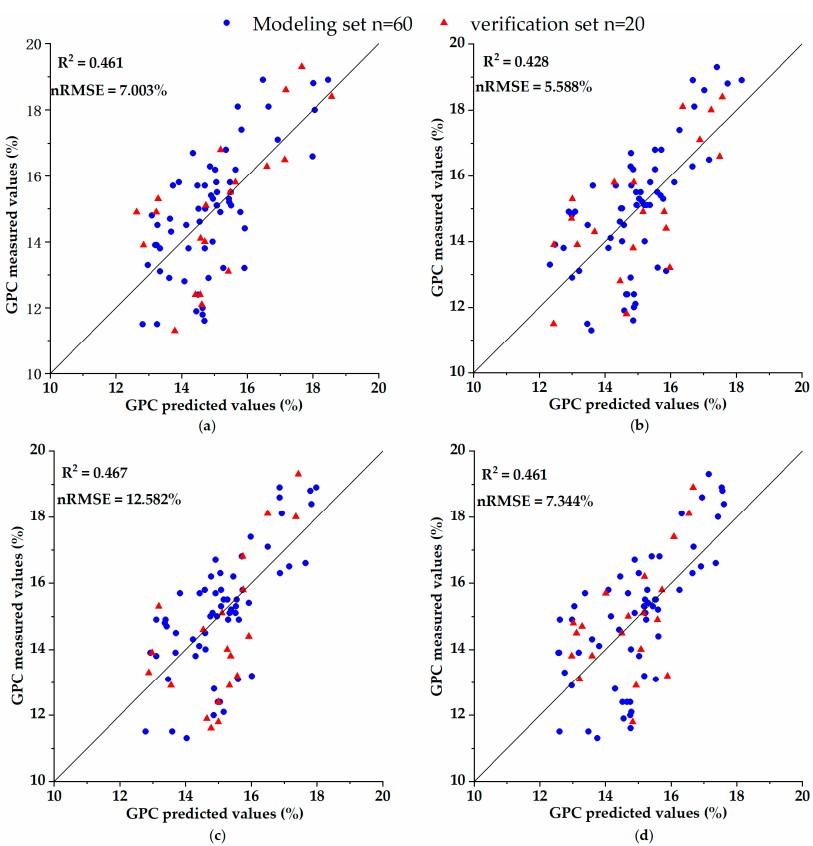

**Figure 8.** Relationship between measured and predicted grain protein content. (**a**) GPC estimation results based on $LNC_{pred}$ and vegetation indices. (**b**) GPC estimation results based on $LNA_{pred}$ and vegetation indices. (**c**) GPC estimation results based on $PNC_{pred}$ and vegetation indices. (**d**) GPC estimation results based on $PNA_{pred}$ and vegetation indices.

The winter wheat GPC has a wide range 11.30% to 19.30%, which is conducive to establishing the protein prediction models for each nitrogen parameter. Comparing the accuracy evaluation for the four nitrogen parameter models in the winter wheat anthesis stage, the maximum $R^2$ was for the PNC-GPC model (0.467) and the minimum $R^2$ was for the LNA-GPC model (0.428). The modeling accuracy of all four models reached a high significant level of difference. According to verify indicator nRMSE based on verification set, the minimum was found in the LNA-GPC model (5.588%), which is not significantly different, and the nRMSE of the LNC-GPC and PNA-GPC models was 7.003% and 7.344%, respectively, which are not significant differences, while the nRMSE for the PNC-GPC model was 12.582%. Therefore, the above four nitrogen parameters can be used to establish inversion models of the winter wheat GPC. Among these four inversion models, PNA and LNC combined with the spectral parameters have highest modeling accuracy $R^2$ and a smaller nRMSE, and so the model has high reliability.

### 3.4. Model Verification

#### 3.4.1. Model Verification by ASD Data Collected from Beijing Suburb

We validated the wheat PNA, PNC, LNA, LNC, and wheat GPC estimation models (Tables 7 and 9) using the experimental data collected in the Beijing suburbs during the winter wheat anthesis stage from 2003 to 2006. There were in total 63 sample data, including the wheat canopy spectrum data, plant nitrogen data, and GPC data. The predicted four wheat nitrogen parameters (PNA, PNC, LNA, and LNC) which were estimated by the resampling of Sentinel-2B VIs as well as the predicted wheat GPC, and which were detected by the nitrogen parameters combined with the VIs were compared with the measured data, and the verify indicator nRMSE was used to verify the accuracy of the models. The results are shown in Table 10.

**Table 10.** nRMSE of the four nitrogen nutrition parameters and wheat grain protein inversion results based on the 2003–2006 experiment.

| Parameters | nRMSE | Parameters | nRMSE |
|------------|-------|------------|-------|
| PNA | 0.264 | $GPC_{PNA}$ | 0.225 |
| PNC | 0.274 | $GPC_{PNC}$ | 0.188 |
| LNA | 0.263 | $GPC_{LNA}$ | 0.175 |
| LNC | 0.295 | $GPC_{LNC}$ | 0.525 |

Table 10 shows that the nRMSE of the four nitrogen nutrition parameters inversion models are all in the range of 26.333–29.53%. While the nRMSE of the grain protein content inversion models inversed by PNC and LNA were 0.188 and 0.175, respectively, the nRMSE models inversed by PNA was 0.225, and the model inversed by LNC was 0.525.

#### 3.4.2. Model Verification by Simulated Sentinel-2B Image by UAV Hyper-Spectral Data

We also validated the wheat PNA, PNC, LNA, and LNC estimation models (Table 7) using the 2014–2015 trial data collected in the National Experimental Station for Precision Agriculture during the winter wheat anthesis stage. There were in total 48 sample data, including the low altitude wheat UAV hyperspectral image and plant biomass and nitrogen data. In this study, the UAV hyperspectral image from Trial 3 was first convolved to the spectral band configuration of Sentinel-2 using the spectral response function of Sentinel-2B, and then, the original 0.02 m simulated sentinel-2 image was resized to 0.5 m for wheat biomass data and four wheat nitrogen parameters (PNA, PNC, LNA, and LNC) were analyzed and calculated based on the wheat plants collected from a 40 × 50 cm subplot within each field. Figures 9 and 10 shows the inversion results of four nitrogen indicators and GPC on 0.5 m UAV data respectively.

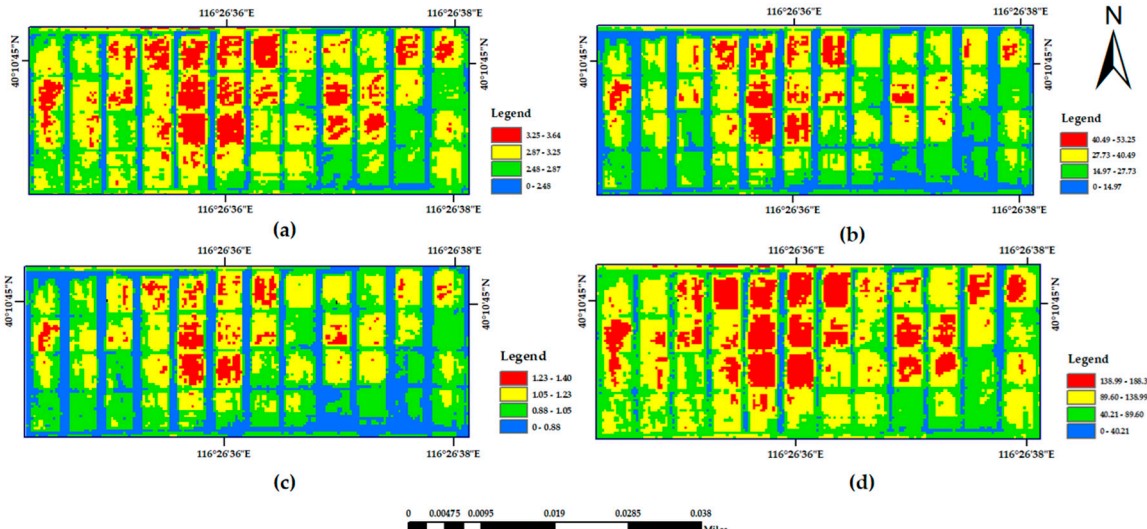

**Figure 9.** UAV data inversion map of four nitrogen nutrition parameters. (**a**) LNC spatial distribution map. (**b**) LNA spatial distribution map. (**c**) PNC spatial distribution map. (**d**) PNA spatial distribution map.

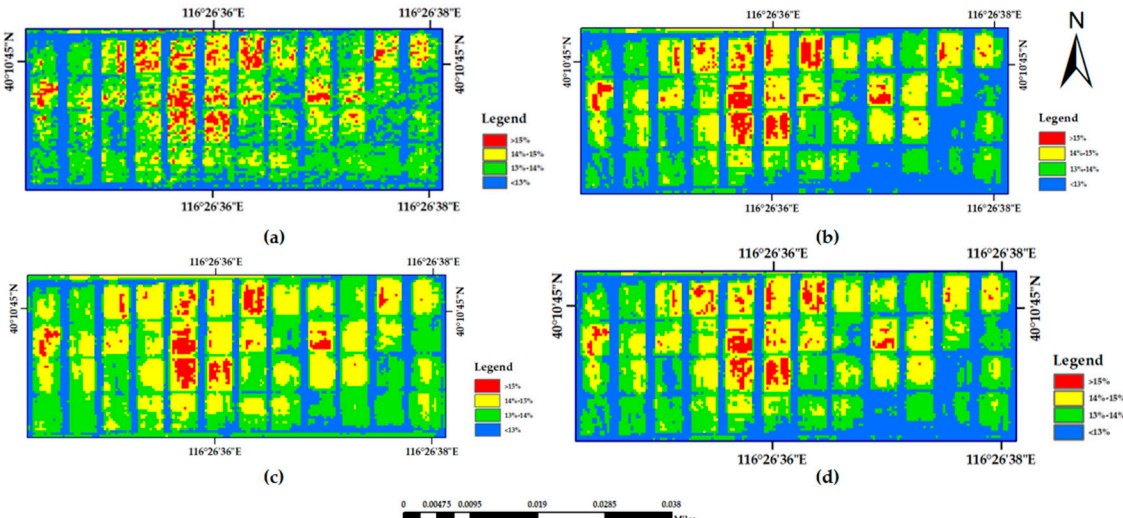

**Figure 10.** UAV data inversion of the grain protein content. (**a**) $GPC_{LNC}$ spatial distribution map. (**b**) $GPC_{LNA}$ spatial distribution map. (**c**) $GPC_{PNC}$ spatial distribution map. (**d**) $GPC_{PNA}$ spatial distribution map.

Hu and Islam's studies revealed that remote sensing model scale invariant assumption is appropriate under two conditions. First, if the parameters are homogeneous over the pixel scale, then the algorithm is scale invariant. Second, if the algorithm can be described as a linear combination of inputs and parameters, then the algorithm is scale invariant [70]. Thus for homogeneous land surface or linear algorithms, remote sensing algorithms can be scaled up or down without any error [71].

In this study, the constructed wheat nitrogen and GPC estimation models are multi-linear models; while Figure 6 also indicates that there are good linear relationship between the nitrogen parameters and the selected vegetation indices. The original 0.02 m simulated sentinel-2 image was also resized to 1 m and 2.5 m spatial resolution images in order to estimate the wheat N and GPC model accuracy. The indicator nRMSE was used to verify the accuracy of the predicted four wheat nitrogen parameters (PNA, PNC, LNA, and LNC) and GPC models which were estimated by the VIs calculated by UAV data in different spatial resolutions. The results are shown in Table 11.

Table 11 shows that UAV- data in different resolutions all have high accuracy in the inversion results of the four nitrogen nutrition indexes as well as grain protein content. The accuracy of the inversion results generally shows a trend of slowing down from 0.50m to 2.50 m resolution resized

data, overall the inversion results of four nitrogen nutrition parameters, PNA and LNC have higher accuracy (0.200 to 0.209, 0.169 to 0.172) than PNC and LNA (0.373 to 0.378, 0.306 to 0.322).

**Table 11.** Normalized root mean square error (nRMSE) of four nitrogen nutrition parameter models based on the different spatial resolution UAV images.

| Parameters | nRMSE | | | Parameters | nRMSE | | |
|---|---|---|---|---|---|---|---|
| | 0.50 m | 1.00 m | 2.50 m | | 0.50 m | 1.00 m | 2.50 cm |
| PNA | 0.200 | 0.200 | 0.209 | $GPC_{PNA}$ | 0.126 | 0.123 | 0.126 |
| PNC | 0.373 | 0.373 | 0.378 | $GPC_{PNC}$ | 0.132 | 0.132 | 0.132 |
| LNA | 0.306 | 0.306 | 0.322 | $GPC_{LNA}$ | 0.127 | 0.127 | 0.127 |
| LNC | 0.169 | 0.169 | 0.172 | $GPC_{LNC}$ | 0.128 | 0.128 | 0.125 |

### 3.4.3. Model Verification by Sentinel-2B Image

Applying the models to the Sentinel-2A image of 8 May 2018, in Renqiu, the following inversion maps of the four different nitrogen indicators during the anthesis stage of wheat were obtained (Figure 11). Applying the model to the Sentinel-2A image of 8 May 2018, in the Renqiu area, the GPC inversion map based on four different nitrogen indicators during the wheat anthesis stage shown in Figure 12 was obtained.

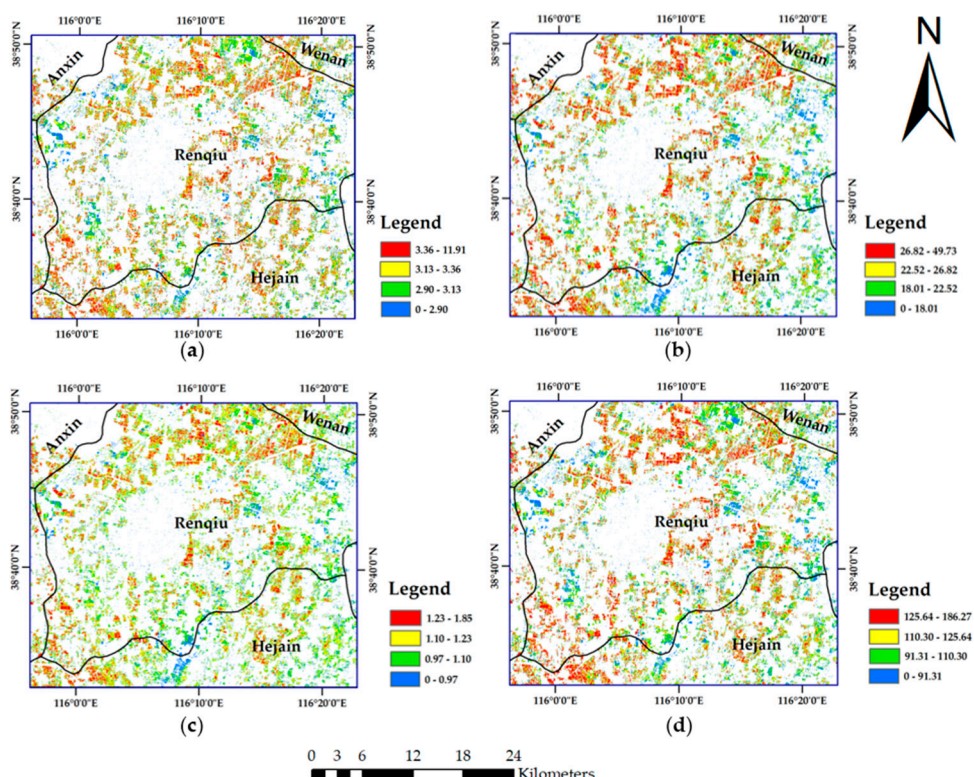

**Figure 11.** Spatial distribution map of the four nitrogen nutrition indexes. (**a**) LNC spatial distribution map. (**b**) LNA spatial distribution map. (**c**) PNC spatial distribution map. (**d**) PNA spatial distribution map.

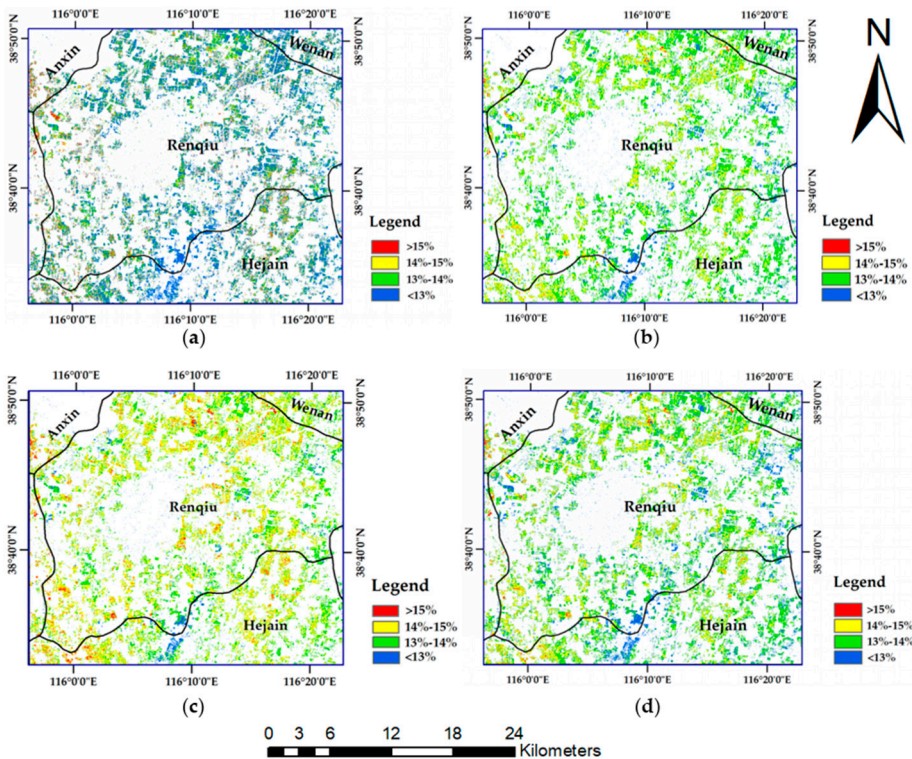

**Figure 12.** Spatial distribution of the grain protein content. (**a**) $GPC_{LNC}$ spatial distribution map. (**b**) $GPC_{LNA}$ spatial distribution map. (**c**) $GPC_{PNC}$ spatial distribution map. (**d**) $GPC_{PNA}$ spatial distribution map.

The 2017–2018 experiment obtained wheat nitrogen data from 20 farmlands in the wheat anthesis stage in the Renqiu area and also obtained the final grain protein quality data of all the study fields. The above wheat nitrogen parameter inversion models (Table 8) were evaluated by the four measured nitrogen nutrition parameters of 20 samples, and four wheat GPC models (Table 9) were also evaluated using the measured wheat GPC data. The accuracy of the models was verified by nRMSE, and the results are shown in Table 12.

**Table 12.** nRMSE of four nitrogen nutrition parameters and wheat grain protein inversion results based on the 2017–2018 experiment.

| Parameters | nRMSE | Parameters | nRMSE |
|---|---|---|---|
| PNA | 0.250 | $GPC_{PNA}$ | 0.795 |
| PNC | 0.428 | $GPC_{PNC}$ | 0.789 |
| LNA | 0.426 | $GPC_{LNA}$ | 0.816 |
| LNC | 0.232 | $GPC_{LNC}$ | 0.125 |

From Table 12, PNA and LNC have the highest accuracy in the inversion results of the four nitrogen nutrition indexes (0.250 and 0.232, respectively), while the PNC and LNA have a lower accuracy of 0.428 and 0.426, respectively. The inversion results of the wheat grain protein content show that PNA, PNC, and LNA have higher accuracy (0.795, 0.789, and 0.816, respectively) and were not significantly different. The accuracy of LNC was 0.125.

## 4. Discussion

Grain protein content is determined by cultivar selection, fertilization, irrigation, and environmental factors [72–77]. However, the main factor in determining the grain protein content may be the product of the nitrogen accumulation at the anthesis stage and the nitrogen transfer efficiency

to the grain [3]. This study revealed that the wheat nitrogen parameters at the anthesis stage and the wheat GPC under strict nitrogen control and field management show a significant relationship. The correlation coefficients between wheat PNA, PNC, LNA, LNC, and GPC were all very significant ($p < 0.01$). These results coincide with the previous study by Zhenhai Li et al. [78]. Our study also indicated that correlation coefficients between the wheat PNA, PNC, LNA, LNC, and the wheat GPC for experiments in the Beijing suburbs from Trials 3–6 were −0.145, 0.240, 0.314, and −0.142, respectively. Apparently, there are many factors that influence the nitrogen transfer efficiency from leaf to grain during the grain filling stage, such as field irrigation and environmental factors, including the moisture or heat stress across a large area.

Our study also revealed that the correlation coefficients between the spectrum VIs and wheat nitrogen parameters collected in the National Experiment Station for Precision Agriculture (Trial 1 and Trial 2; Table 5) were more significant than those collected in the Beijing suburbs in Trials 4–7. The possible reasons for this phenomenon are that wheat canopy spectrum data collected in a farm were controlled within 1–2 h, so the spectrum difference caused by variations in the Sun elevation angle, Sun azimuth, and atmospheric conditions was limited in the experiment. In contrast, it took more than 5 h when the spectrum data were collected in the experiment fields located in Beijing suburbs; the atmospheric conditions varied very quickly when the researchers moved from one field to another field. Although the wheat canopy spectrum calibration was performed before and after the measurements using a calibration plate, the spectrum difference caused by atmospheric conditions apparently affected the precision of wheat nitrogen detection by the spectrum data.

This study reveals that there will be more challenges for detecting the cereal nitrogen status and grain quality monitoring at the regional scale through field canopy remote sensing techniques. The wheat plant nitrogen status and GPC are affected by many factors, such as soil nutrition, weather conditions, and field management. These factors, along with cultivars, contribute to the spatial variability of the crop nitrogen status and GPC. Satellite imaging may be helpful to monitor the crop growth and to predict the wheat GPC for large areas and to untangle the aforementioned factors.

Accumulation of plant nitrogen is the only direct resource for grain protein, which forms when nitrogen is physically transferred into grains at the grain filling stage. Such a relationship implies a correlation between the plant nitrogen content and the grain protein content. The results from this study demonstrate that the grain protein content was positively correlated to the leaf nitrogen content at the anthesis stage at the 99.9% significance level. The method that indirectly establishes the relationship between the spectral parameters and the grain protein content through agronomic parameters as an intermediate variable is more mechanistic, and the model has higher stability and scalability, which has become a research hotspot in recent years [79]. As a bridge connecting the spectral parameters and the grain protein content, agronomic parameters must not only reflect the level of grain protein content but also have a significant correlation with spectral parameters. Chemura et al. [80] have assessed the feasibility of Sentinel-2 MSI spectral bands and vegetation indices in empirical estimation of coffee foliar N content at landscape level with Sentinel-2 data; results showed that coffee foliar N is related to Sentinel-2 MSI B4, B6, B7, B8 and B12 bands, and relative vegetation indices were more related to coffee foliar N, combining optimized bands and vegetation indices produced the best results in coffee foliar N modelling ($R^2 = 0.78$, RMSE = 0.23). Since the quality of wheat is determined by many factors, considering the correlation between the spectral parameters and the grain content of wheat protein in previous research, in this study, the spectral parameters were added as the inversion parameters in the process of inverting the grain protein content combined with agronomic parameters in order to improve the inversion accuracy of the grain protein content. The result shows that the $R^2$ of the multi-linear regression inversion models of grain protein quality were up to 0.467 and the minimum nRMSE was 7.003%, which shows that the inversion model has high precision and reliability.

Previous studies on the prediction of grain protein content in a large area using satellite data are rare and most of the related studies are based on the spectral parameters/grain protein content model. Changwei Tan et al. [81] analyzed the quantitative relationship between satellite remote sensing

variables and winter wheat grain protein content and constructed an inversion model of winter wheat grain protein content based on the multi-vegetation index using the partial least squares method and Landsat TM images. The root mean square error (RMSE) of the model was 0.37% and the coefficient of determination ($R^2$) was 0.642, and thus the inversion effect was ideal. This paper attempted to apply the inversion model of spectral parameters/agronomic parameters/grain protein content to Sentinel-2A/2B images to achieve a wide-range remoting sensing prediction of nitrogen nutrition parameters and grain protein content. The results show that the complex model with agronomic parameters still has high precision and reliability, and the mechanism and stability of the model were greatly increased.

There are still some shortcomings in this study. Because of factors such as atmospheric aerosols and different sensor types, there is a certain deviation between the reflectance value of the simulated Sentinel-2A/2B data and the actual Sentinel-2A/2B image data, which is one of the main reasons for the error in the inversion results when using the model to invert the satellite data.

Previous study indicates that for homogeneous land surface or linear algorithms, remote sensing algorithms can be scaled up or down without any error. In this study, although the winter wheat nitrogen nutrition parameters are estimated by near-linear models, some vegetation indices used in those models are calculated by nonlinear algorithm (Table 4). There must be subpixel scale heterogeneity effects when using those models to estimate wheat nitrogen status through UAV or Sentinel 2 images. We will analyze the effects caused by the VIs and seek to utilize proper approach to correct spatial scale effects in the future studies.

In addition, in 2018, there were inconsistencies in the anthesis stage of the 20 plots in the Renqiu area. The actual farmland wheat sampling date was from 5 May to 10 May. There is a difference between the wheat nitrogen information of the relevant sample and the satellite acquisition date (8 May 2018), which also led to an incomplete correspondence between the satellite spectral information and the crop nitrogen information, which affected the final GPC inversion results. Therefore, eliminating the impacts of cultivation, climate, and other factors in different plots in the large area is also a problem that needs to be solved in further studies for the development of wheat nitrogen and GPC remote sensing at a regional scale.

## 5. Conclusions

In this paper, wheat canopy hyperspectral data were resampled to simulated Sentinel-2A VIs and the relationship between the VIs and the four nitrogen parameters (PNA, PNC, LNA, and LNC) in the winter wheat anthesis stage and wheat GPC were analyzed. A wheat GPC estimation model was constructed that combined the sensitive VIs and wheat nitrogen parameters through a multiple linear regression algorithm. The models were then verified by the simulated Sentinel-2A data across different years and real Sentinel-2A images of a county area. The results of this study indicate that:

(1) By analyzing the correlation between the Sentinel-2A VIs and the wheat nitrogen parameters or wheat GPC, the most significant relationship between wheat nitrogen parameters and GPC was found in the farm-scale experiment. The correlation coefficients for Sentinel-2A VIs and wheat nitrogen parameters also reached a very significant level in the wheat anthesis stage, which provides the potential for the estimation of wheat GPC through spectral VIs and the wheat nitrogen parameters.

(2) A total of four nitrogen parameter estimation models were established using the simulated Sentinel-2A multi-vegetation index through the MLR algorithm and all have high modeling accuracy and verification accuracy. Among them, the PNA modeling had the highest accuracy and reliability with the calibration $R^2$ of 0.807 and verification nRMSE of 13.940%.

(3) The wheat GPC was predicted by the four nitrogen parameters combined with the spectral parameters and the inversion model based on PNA, IREC, and CVI was the most accurate and reliable model ($R^2$ = 0.461, nRMSE = 7.344%).

(4)  We verified the accuracy of relevant models using ground-measured data obtained from 2003–2006 experiments in the Beijing suburbs. The prediction results of the four nitrogen nutrition parameters all showed an acceptable accuracy while the prediction results of the GPC, PNC, and LNA showed a good accuracy and PNA showed an acceptable accuracy. These three nitrogen nutrition parameters can better invert the grain protein content of wheat, which indicates that the models had good inter-annual and inter-regional expansion.

(5)  Applying the relevant models to the Sentinel-2A imagery obtained in Renqiu county in 2018 indicated that the nRMSE of PNA and LNC were 25.241% and 23.200%, respectively. The nRMSE for the GPC models based on PNA, PNC, and LNA and VIs were 8.040%, 7.888%, and 8.162%, respectively, which is not different. The nRMSE of the LNC was 12.461%. Based on the results of all the inversions, the model with PNA as the intermediate variable is a relatively reliable choice for the inversion of satellite image data.

**Author Contributions:** H.Z. and X.S. processed and analyzed the data and drafted the manuscript. X.S. guided the experimental design, participated in data collection, advised on data analysis, and revised the manuscript. G.Y., Z.L., D.Z., H.F., and C.Y. were involved in the experiments, ground data collection, and manuscript revision. All authors read and approved the final version.

**Funding:** This work was supported by the National Key Technologies of Research and Development Program (2016YFD0300603), (2016YFD070030303) and National Natural Science Foundation of China (41701375).

**Acknowledgments:** We appreciate the help from Hong Chang and Weiguo Li during field data collection. The funders had no role in choosing the study design or in the collection, analysis, and interpretation of the data, in the writing of the report, or in the decision to submit the article for publication.

**Conflicts of Interest:** The authors declare no conflict of interest.

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
