# Peer review of "Monitoring of Nitrogen and Grain Protein Content in Winter Wheat Based on Sentinel-2A Data"

_remotesensing, doi:10.3390/rs11141724_

Round 1

Reviewer 1 Report

Authors have improved the manuscript and it can now be accepted for publication.

Reviewer 2 Report

Although the authors have not fully replied to the question of upscaling from UAV to Sentinel-2, the paper has clear methodological and experimental contents to deserve its publication

Reviewer 3 Report

The research article “Monitoring of Nitrogen and Grain Protein Content in Winter Wheat Based on Sentinel-2A Data” is a great study on using remotely sensed data, in this case Sentinel-2A imagery data, for modeling grain protein content. Based on the red highlighted context, the authors have made a very comprehensive revision based on suggestions and comments, the context is much better shape now. So this version should be accepted as it is.

This manuscript is a resubmission of an earlier submission. The following is a list of the peer review reports and author responses from that submission.

Round 1

Reviewer 1 Report

The aim of the the research was to develop a remote sensing framework to estimate wheat N and GPS using S2 derived Vis for field monitoring and early prediction of wheat quality. The research is well implemented and well written and can be accepted if the authors address the following comments. .

Comments

Abstract: Line 14: “GPC based on remote sensing is of great significance for agricultural production” Revise for clarity

Line 28-29: What does “ medium difference mean? Revise.

Line 42: Replace “safety” of agricultural product with another word.

Line 62: precision farming is different from precise farming. Use precision farming.

Line 106-115: There is need for justification for the application of Sentinel by referring to studies that have used Sentinel in N estimation. This will strengthen their hypothesis that it can perform the task.

Line 116: replace estimate with investigate

Line 126: Before experimental design describe the study area, so line 127-144 will be in this section with the map.

Line 145-146: better to stick to hectares as they are equivalent to hm-2 but are more standard.

Line 150: what is the meaning of “other” which was according to actual management? A reference to standard practices manual or elsewhere where they have been described will suffice.

Line 227: Give reference for ENVI software.

Line 248-249: The resampling process as presented is not represented of the response function of s2 bands because it assumes maximum reflectivity for all band numbers between the start and the end. The Full Width at Half-Maximum (FWHM) would have been best and since they used ENVI, there is already a built-in function for that. This will not significantly improve their results but is obviously a better way for resampling.

Line 256: The S2 response function figure is not necessary as this is available already . Just a reference to it will be sufficient.

Line 282-284: Give a reference for the descriptions of the nRMSE into the given bins.

All methods: A flow chart to summarize the study is necessary as there are many parts to the study.

Line 285: Results is more standard

Line 303: How do you define 0.4 as the level of significance or threshold? References can help.

Line 307, replace very with highly significant.

Line 355: The predicted vs observed plots should be inverted (read Pineiro et al 2013, Ecol.Modelling, 216, 316-322). There is need also to make sure that the units on both the x and y-axis have same range and limits for a fair comparison. A 1:1 line can also be helpful.

Line 368: The bins for th4e maps should be equal for comparison. It is also better to remove the 0 as non-wheat as it is not true that it is actually 0. Also add a-d for easier image referencing.

Figure 7: Make sure the plots are square.

Discussion: Authors should discuss their results relative to other studies on N mapping and estimation with Sentinel. They are overly optimistic with the high nRMSE levels and may need to be discussed more objectively.

Author Response

Dear reviewer:

Thank you very much for your comments about our paper submitted to Remote Sensing. We have carefully addressed and incorporated the comments and suggestions in the revised manuscript and we have provided a point-by-point response to your comments in the upload Word file. We appreciate the time and efforts put forth by you to help us publish this information. These comments and suggestions have greatly increased the clarity and strength of the manuscript.

Reviewer 2 Report

This paper aims at defining empirically the nitrogen and grain protein content on the basis of field spectral measurement, transferred to Sentinel-2 observations. The empirical retrieval is based on statistical regressions between several spectral indexes and wheat parameters determined by means of destructive sampling.

Whilst the objectives of the paper are clear, the sequence of the experiment is described in a cumbersome way. The field experiments in Table 1 are not numerated chronically. It seems that Trails from 3 to 6 were conducted with a different purpose under uniform management conditions, and at a later stage the Trials 1 and 2 were carried out to derive the empirical relationships between canopy parameters and spectral indexes. The consistency between the different trials is not clear, or it not clearly written in the text. It seems that also the size of the experimental plot in each trial is different. How does this impact the calibration and validation?

A more relevant issue concerning this paper is that there is no reference about the values of Leaf Area Index for the different plots during the experiment. The scale of measurements on leaves samples and field spectroscopy might have the same spatial scale, but then to transfer the empirical relationships at canopy level or plot scale (as in the case of Sentinel-2 application) it would be needed to take into account the amount of foliage on the considered footprint. In the paper different measurement scales are considered, but the analysis of experimental data is carried out regardless of the spatial extent.

Author Response

(The authors gave the same response as above.)

Reviewer 3 Report

I think the paper is well written and interesting. Moreover the three locations and the trials repeated in more years give value to your work. There are very few suggestions and comments. I suggest to see the comments about some of the tables presented. 

Author Response

(The authors gave the same response as above.)

Round 2

Reviewer 2 Report

The response from the Authors is not satisfactory. As evidenced by Eqs.(1) (2) and the text at lines 194-195 was determined for 1 single plant, without considering the IFOV of ground spectral measurements and especially of Sentinel. The extrapolation of empirical relationships based on this approach to Sentinel is not correct.